# Flex-Forcing: Towards a Unified Autoregressive and Bidirectional Video Diffusion Model

**Xinyin Ma** [1 2]  **Julius Berner** [2]  **Chao Liu** [2]  **Arash Vahdat** [2]  **Weili Nie** [2 *]  **Xinchao Wang** [1 *]

## Abstract

Recent progress in large-scale generative models has substantially advanced video generation, yet existing methods remain constrained by a rigid inference paradigm. Bidirectional diffusion models excel at global coherence and visual fidelity but suffer from slow inference, while autoregressive models offer efficient and streaming generation at the cost of long-range consistency and exposure bias. We introduce Flex-Forcing, a unified training and inference framework that enables a video diffusion model to seamlessly operate under both bidirectional and autoregressive generation regimes. The core idea is a flexible chunking mechanism jointly defined over the temporal axis and denoising steps. This design allows the model to (1) perform flexible chunking according to different device budgets, (2) perform bidirectional inference across chunks for global structure planning, while generating frames autoregressively within each chunk for efficient and fine-grained synthesis, and (3) perform any-order, any-timestep autoregressive generation without the strict causal constraint. Extensive experiments on multiple video generation benchmarks demonstrate that Flex-Forcing achieves consistently better video quality, long-video stability than strong baselines with a rigid inference schedule, while offering faster inference.

## 1. Introduction

Video generation has recently witnessed rapid progress driven by large-scale generative models (OpenAI, 2024; Kuaishou, 2024; Runway, 2024; Google, 2024), leading to

---
[*]Equal Advising  [1]Electrical and Computer Engineering, National University of Singapore, Singapore  [2]NVIDIA Research, California, USA. Correspondence to: Xinchao Wang <xinchao@nus.edu.sg>, Arash Vahdat <avahdat@nvidia.com>.

*Proceedings of the 43$^{rd}$ International Conference on Machine Learning*, Seoul, South Korea. PMLR 306, 2026. Copyright 2026 by the author(s).

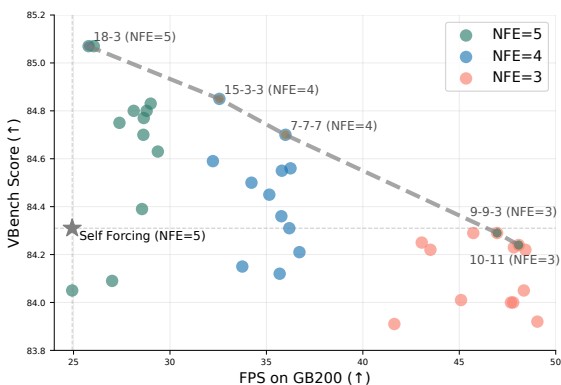

*Figure 1.* In various configurations, our method exhibits superior efficiency and performance compared to self-forcing, measured by FPS (frames per second) and VBench score.

substantial improvements in visual realism, temporal coherence and semantic consistency (Yang et al., 2024; Wan et al., 2025; Wu et al., 2025; Chen et al., 2025b; HaCohen et al., 2026). These advances have enabled a wide range of emerging applications, including long-form video synthesis (Chen et al., 2025a; Yang et al., 2025), interactive world modeling (He et al., 2025; Hong et al., 2025), and creative intelligence systems (Hu, 2024; Jiang et al., 2025), where models are required to generate temporally extended, content-rich sequences with fine-grained control. As video models continue to scale in both capacity and context length, the efficiency and flexibility of their generative mechanisms have become increasingly critical, particularly for scenarios that demand customized and flexible trade-offs between quality, latency, and computational cost.

Among existing video generation approaches, two dominant paradigms have emerged: the bidirectional diffusion paradigm, widely adopted in pretrained video diffusion models (Ho et al., 2022b; Hong et al., 2022; Chen et al., 2024b), and the autoregressive paradigm, which has recently emerged as an efficient alternative (Deng et al., 2024; Wang et al., 2024; Teng et al., 2025; Yin et al., 2025; Huang et al., 2025a), each offering complementary advantages. The bidirectional paradigm leverages full-context attention to jointly model all frames, resulting in strong temporal coherence and high visual fidelity (Ho et al., 2022b), and is therefore well-suited for capturing long-range dependencies such

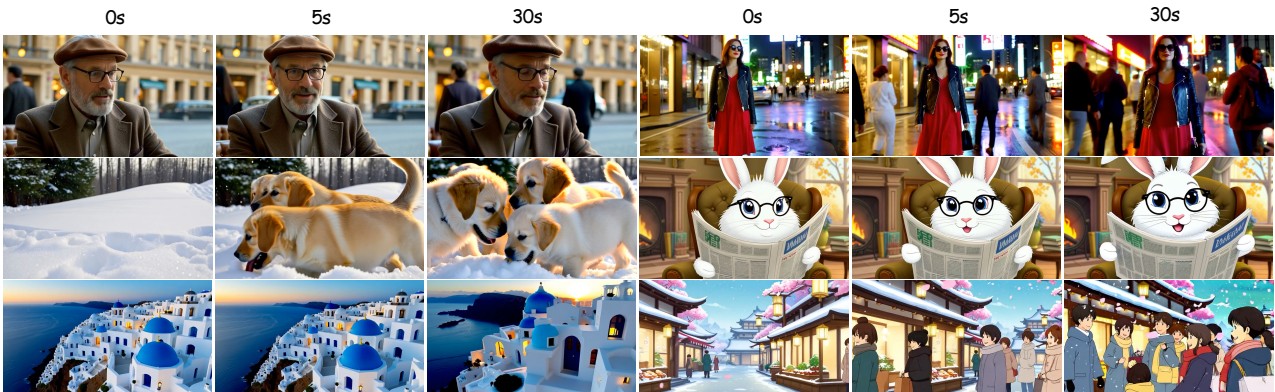

*Figure 2.* Generated examples from Flex-Forcing. We show three frames from the 30s 480p video under six different prompts.

as camera motion, scene transitions, and complex interactions (Yin et al., 2023; Singer et al., 2022). However, the bidirectional paradigm involves substantial computational overhead, leading to high inference costs, which limits its applicability to real-time settings and its scalability to long video generation (Yin et al., 2025; Yu et al., 2023). In contrast, autoregressive video generation models operate under causal conditioning, generating frames sequentially while reusing past key–value states via KV caching. This enables real-time inference and naturally supports arbitrary-length video generation without reprocessing earlier frames (Hong et al., 2022; Villegas et al., 2022). Nevertheless, the lack of global context during generation makes autoregressive models susceptible to error accumulation over time, resulting in drift in object appearance, motion, or scene structure, as well as weaker global temporal consistency and limited long-horizon planning ability (Liu et al., 2025).

Despite their complementary strengths, existing work lacks an effective mechanism to jointly realize the advantages of bidirectional and autoregressive video generation paradigms within a unified framework, and consequently optimizes for one paradigm while compromising the other (Yin et al., 2025; Huang et al., 2025a). We therefore seek to unify both paradigms within a single model, enabling coherent video generation while supporting efficient long-video inference without increasing exposure bias or computational cost. To this end, we propose Flex-Forcing, a unified framework that enables test-time control over the inference paradigm. A single trained model can flexibly operate in (1) a bidirectional mode for globally consistent generation, (2) an autoregressive mode for scalable long-video or streaming synthesis, or (3) a semi-autoregressive hybrid mode that provides intermediate trade-offs between quality and efficiency under different deployment constraints.

To enable such a flexible inference behavior, the core challenge in training is to equip a single model with both causal (autoregressive) and non-causal (bidirectional) generation capabilities. This allows the model to operate in intermedi-

ate regimes between strictly causal and fully bidirectional inference. We address this challenge by introducing *Flexible Chunking*, defined along two orthogonal axes, temporal frames and denoising steps, under which autoregressive and bidirectional inference emerge as two extreme cases. This formulation is consistently applied into both training and inference, which exposes the model to mixed causal and non-causal conditioning contexts. The same query token may attend to key-value pairs of different noise levels, where causal past tokens are typically more denoised, while non-causal future tokens are noisier. To reconcile this noise-level discrepancy, we introduce an alignment mechanism that explicitly enforces representation consistency between causal and non-causal attention contexts.

Our results demonstrate that bidirectional and autoregressive inference can be unified within a single Flex-Forcing model, rather than treated as mutually exclusive paradigms. The resulting flexible inference mode achieves a strong quality–efficiency trade-off, yielding a markedly improved Pareto frontier (Figure 1) and substantially outperforming Self-Forcing (Huang et al., 2025a) across both short- and long-video benchmarks. Beyond generation, this flexible paradigm enables downstream applications with adaptive causal constraints, such as *any-order, any-step autoregressive editing*, which supports localized temporal edits while preserving global video coherence with the original video.

To summarize, our contributions are:

- We propose *Flex-Forcing*, a unified framework that supports both bidirectional and autoregressive video generation within a single model.

- We introduce *flexible chunking* along the temporal and denoising axes, together with a *train–test consistent objective*, and an *aligned conditioning* mechanism to bridge different causal contexts.

- Extensive experiments show that Flex-Forcing substantially outperforms causal models on the performance-

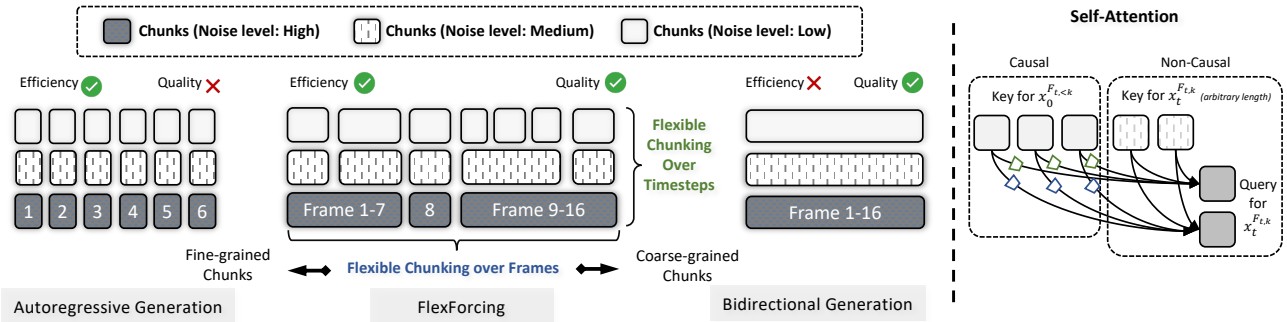

*Figure 3.* (Left) Flexible chunking for bridging the autoregressive and bidirectional video generation. Flex-Forcing adjusts chunk granularity across noise levels while a unified self-attention mechanism supports both causal and bidirectional inference. (Right) The mixed attention with causal tokens and non-causal tokens. We add a timestep dependent K-Projection at the clean cache from past frames.

efficiency trade-off while achieving competitive performance with bidirectional models.

- We further introduce *any-order, any-step autoregressive editing* based on Flex-Forcing, which enables controllable autoregressive editing of videos.

## 2. Related Work

**Bidirectional models for video generation.** Early diffusion-based video generation models extend image diffusion models (Dhariwal & Nichol, 2021; Rombach et al., 2022; Karras et al., 2022; Black Forest Labs, 2024; Esser et al., 2024) to the spatio-temporal domain by jointly denoising all frames with full temporal context (Ho et al., 2022b; Bar-Tal et al., 2024; Luo et al., 2023). These models typically adopt bidirectional self-attention over the temporal axis, allowing each frame to attend to both past and future frames during denoising (Ho et al., 2022a). Subsequent works scale this paradigm using transformer-based backbones and large-scale video–text datasets, establishing bidirectional diffusion models as foundation architectures for general-purpose text-to-video generation (Yang et al., 2024; Blattmann et al., 2023b;a; Gupta et al., 2024).

**Autoregressive models for video generation.** Autoregressive video generation models generate videos by sequentially predicting frames, enabling real-time generation, prompt control and long video generation (Teng et al., 2025). VideoGPT (Yan et al., 2021) uses a GPT-like model to autoregressively model discrete latents using spatio-temporal position encodings. Pyramidal flow (Jin et al., 2024) crafts autoregressive video generation with a temporal pyramid to compress the full-resolution history . Lumos-1 (Yuan et al., 2025) introduces autoregressive discrete diffusion forcing to mitigate the frame-wise imbalance loss. Besides training native autoregressive video generation models, another line of work explores causal distillation (Yin et al., 2025; Huang et al., 2025a; Cui et al., 2025; Liu et al., 2025), which use diffusion forcing (Chen et al., 2024a) and causal attention

mask to initialize the model and then use DMD (Yin et al., 2024b;a) and self-rollout (Huang et al., 2025a) in training.

## 3. Flex-Forcing

**Overview.** We propose a unified generative framework that supports both bidirectional and autoregressive inference. To the best of our knowledge, this is the first approach to demonstrate that these two paradigms can inherently co-exist within a single trained model. The core idea is a flexible chunking strategy that composes chunks of varying granularity. Flexible chunking is learned by post-training a bidirectional video diffusion model and is defined along two orthogonal axes: (i) the video-frame axis, which allows heterogeneous chunk sizes across different temporal regions; and (ii) the denoising-timestep axis, which allows the chunk granularity to change throughout the denoising trajectory. Together, they induce a rich configuration space and enable the model to seamlessly operate in bidirectional or autoregressive modes for quality–efficiency trade-offs.

### 3.1. Flexible Chunking over Video Frames

We represent a video as a sequence of $F$ frames $x = (x^{(1)}, x^{(2)}, \ldots, x^{(F)})$. Diffusion sampling performs $T$ denoising steps with $\{x_t\}_{t=1}^T$. A chunk consists of a consecutive sequence of frames and we make those chunks to be flexible in size. In this way, we have a hybrid factorization in which intra-chunk dependencies are modeled bidirectionally, while inter-chunk generation remains autoregressive. Specifically, at each denoising step $t$, we define a contiguous partition of frame indices $\{1, \ldots, F\}$ by chunk boundary indices $\mathbf{a}_t = (a_{t,0}, a_{t,1}, \ldots, a_{t,K_t})$, where $1 = a_{t,0} < \cdots < a_{t,K_t} = F + 1$. The $k$-th chunk at timestep $t$ covers the frame index set:

$$\mathcal{F}_{t,k} \triangleq \{ f \mid a_{t,k-1} \leq f < a_{t,k} \}, \tag{1}$$

At each denoising step $t$, given a partition $\mathbf{a}_t$, the chunk-wise causal sampling now is defined as:

$$x_{t-1}^{\mathcal{F}_{t,k}} \sim q_\theta\left(x_{t-1}^{\mathcal{F}_{t,k}} \mid x_0^{\mathcal{F}_{t,<k}}, x_t^{\mathcal{F}_{t,k}}; \mathbf{a}_t\right), \qquad (2)$$

where $\mathcal{F}_{t,<k} \triangleq \bigcup_{u<k} \mathcal{F}_{t,u}$ denotes all frame indices before current chunk, and $x^{\mathcal{F}}$ denotes the frames in chunk $\mathcal{F}$. Following self-forcing (Huang et al., 2025a), $x_0^{\mathcal{F}_{t,<k}}$ is represented by the KV cache of previously predicted frames in the diffusion transformer (Peebles & Xie, 2023).

Eq. 2 formalizes that later chunks are generated via denoising conditioned on earlier chunks. The above formulation implies a flexible chunking strategy along the video's frame axis. By setting the frame-selection parameter $\mathbf{a}_t$, our framework defines a variable subset of frames that can act as the history under the causal constraint and the context that interacts in the bidirectinal attention. Table 1 provides a unified view of auoregressive and bidirectional inference as specific cases of our hybrid inference.

| Mode | $\mathbf{a}_t$ | $q_\theta(x_{t-1} \mid x_t)$ |
|---|---|---|
| Autoregressive | $(1, 2, 3, \ldots, F, F+1)$ | $\prod_{k=1}^{F} q_\theta\left(x_{t-1}^k \mid x_0^{<k}, x_t^k\right)$ |
| Bidirectional | $(1, F+1)$ | $q_\theta(x_{t-1}^{1:F} \mid x_t^{1:F})$ |
| Hybrid | $(a_{t,0}, a_{t,1}, \ldots, a_{t,K_t})$ | $\prod_{k=1}^{K_t} q_\theta\left(x_{t-1}^{\mathcal{F}_{t,k}} \mid x_0^{\mathcal{F}_{t,<k}}, x_t^{\mathcal{F}_{t,k}}; \mathbf{a}_t\right)$ |

*Table 1.* Unified view of autoregressive, bidirectional, and hybrid inference as specific cases of the boundary configuration $\mathbf{a}_t$.

## 3.2. Flexible Chunking over Denoising Timesteps.

As defined in Eq. 2, the $t$-dependent chunk partition $\mathbf{a}_t$ allows the chunking strategy to vary across denoising timesteps. Intuitively, early denoising steps, characterized by high noise levels, focus on global structure and benefit from larger temporal chunks. Conversely, later refinement steps prioritize local details, which can be effectively modeled with smaller chunks that require less long-range context. This results in a hierarchical, pyramid-like structure where chunk size decreases as noise levels drop (Figure 3).

Formally, recall that $\mathbf{a}_t = (a_{t,0}, a_{t,1}, \ldots, a_{t,K_t})$ denotes the frame partition at denoising timestep $t$. We impose a *nested flexibility* that allows each chunk to be further subdivided as denoising proceeds. Specifically, after completing the denoising at step $t+1$ (i.e., $x_{t+1} \to x_t$), we split the partition for $t$ by *inserting* additional boundaries while preserving all existing ones. Timestep $t$ inherits the same configuration of $t+1$: $\mathcal{F}_{t,k} = [a_{t,k-1}, a_{t,k}) = [a_{t+1,k-1}, a_{t+1,k})$, and we introduce a sequence of new split points $\mathcal{S}_{t,k}$ with the number of sub-chunk boundaries $n_{t,k} \geq 0$. Simply merging the original endpoints and the split points gives the new chunking patterns:

$$\mathbf{a}'_{t,k} = a_{t,k-1} \cup \mathcal{S}_{t,k} \cup a_{t,k}, \qquad (3)$$

This speratability can be applied *recursively* and the resulting sub-chunks can be further split by inserting additional boundaries in the next denoising step.

During inference, frames are processed in temporal order. However, under the pyramid chunking strategy, a synchronization challenge arises when a large chunk at denoising step $t$ is partitioned into smaller sub-chunks at step $t-1$. In these instances, we temporarily buffer the denoising results for all frames within the original bidirectional chunk at step $t$. We then autoregressively resume denoising for each sub-chunk at step $t-1$ once the required KV caches from preceding sub-chunks become available. This execution order ensures that causal dependencies are satisfied across varying granularities (see Figure 13 in Appendix).

## 3.3. Flexible-chunk Training

With the unified formulation in place, we next incorporate it into the training objective. The training consists of two parts: (1) introducing causality constraints into a bidirectional diffusion model while preserving its non-causality ability to attend to future chunks, and (2) enabling the model to attend to key states derived from inputs at different noise levels.

**Injecting the causality when maintaining the non-casuality** We largely follow the training paradigm introduced in CausVid (Yin et al., 2025) and Self-Forcing (Huang et al., 2025a), which transfers a bidirectional diffusion model into a causal one. This training pipeline consists of two stages. (1) ODE initialization, where a causal attention mask is applied to inject causality into the model. (2) Asymmetric distillation with DMD training, which further strengthens causality through self-rollout and applies the VSD loss (Yin et al., 2024b) to distill a multi-step diffusion model into a few-step one. We find that injecting causality while preserving non-causal modeling capacity can be naturally achieved during the asymmetric distillation stage.

In the asymmetric distillation stage, we introduce a stochastic chunking strategy that dynamically varies the attention pattern within each rollout. Recall that we define a contiguous partition of the frame index set via a boundary configuration, we randomly sample a chunk partition $\mathbf{a}_t$ during training, and apply rollout to these flexible chunks in a dynamic way. After the latents of all frames $\mathbf{x}_0 = \{x_0^i\}_{i=1}^F$ has been rolled out under each $\mathbf{a}_t$, we train the generator $G_\theta$ by taking the gradient from (Yin et al., 2024b):

$$\nabla_\theta D_{KL} = \mathbb{E}_{\mathbf{z} \sim \mathcal{N}(0;\mathbf{I})} \left[ -\left(s_{\text{real}}\left(\mathbf{x}_t\right) - s_{\text{fake}}\left(\mathbf{x}_t\right)\right) \frac{\partial G_\theta}{\partial \theta} \right] \qquad (4)$$

where $s_{\text{real}}$ and $s_{\text{fake}}$ denote the real and fake scores, respectively, and $\mathbf{x}_t = I(G_\theta(\mathbf{z}), t)$ with $I(\mathbf{x}_0, t)$ representing a forward process in diffusion models. By randomly sampling the chunk size, the model is exposed to attention configu-

rations that range from strictly causal to fully non-causal. As a result, the training implicitly contains a heterogeneous mixture of causal and bidirectinal dependencies.

**Noise-level alignment across causal and non-causal attention.** Since our model supports both causal and bidirectional inference, the same query token may attend to contexts produced under different generative regimes. At a given self-attention layer, this implies that a query can attend to key–value states derived from more denoised representations (causal past tokens) as well as states generated from noisier latents at higher diffusion timesteps (non-causal future tokens). As a result, the attended KV states exhibit substantially different signal-to-noise ratios. Standard self-attention treats all key–value pairs uniformly, which introduces a noise-level mismatch in context aggregation and degrades performance in flexible inference settings.

To address this issue, we propose a noise-aligned projection for the key states, termed K-Projection. Concretely, it projects the cache for key states generated from clean inputs into the "noisy" latent space corresponding to the current diffusion timestep. After projection, all key states are expressed in a noise-consistent representation space, enabling standard self-attention to be applied. Formally, the timestep-dependent projection is defined as:

$$\Pi_{t \leftarrow 0} : \mathbb{R}^d \to \mathbb{R}^d, \qquad \tilde{K}_t = \Pi_{t \leftarrow 0}(K_0). \tag{5}$$

where $\Pi_{t \leftarrow 0}$ is a lightweight, timestep-conditioned linear projection and is intialized as identity mappings. For each denoising step $t$, the self-attention computation is then performed as:

$$\text{Attn}(Q_t, \tilde{K}_t, V_t) = \text{softmax}\left(\frac{Q_t \tilde{K}_t^T}{\sqrt{d}}\right) V_t$$

$$\text{where } \tilde{K}_t = \text{concat}\left(\Pi_{t \leftarrow 0}\left(K_0^{<\mathcal{F}_{t,k}}\right), \tilde{K}_t^{\mathcal{F}_{t,k}}\right) \tag{6}$$

The projection is applied on-the-fly during attention computation, without modifying the cached KV tensors or gradient propogation on the KV cache. During inference, clean KV states are stored once and dynamically projected according to the diffusion timestep. This design preserves the efficiency benefits of KV caching while enabling stable flexible inference across causal and non-causal regimes.

## 4. Applications of Flex-Forcing

### 4.1. Inference Flexibility: Better Speed, Better Quality

We begin by presenting the most straightforward application of Flex-Forcing, which enables a hybrid and flexible inference paradigm that combines autoregressive and bidirectional inference at test time. By allowing variable chunk sizes, Flex-Forcing supports adaptive and searchable chunk

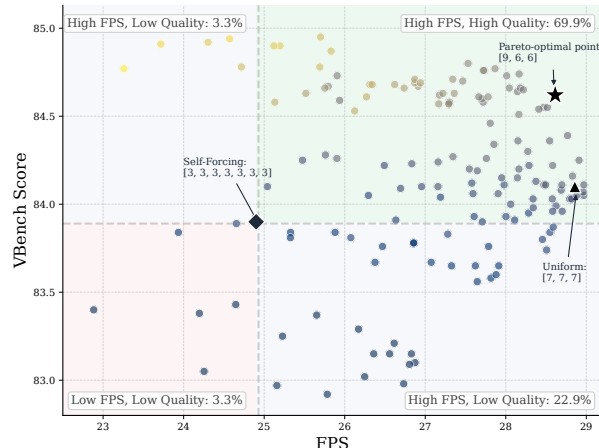

*Figure 4.* We test the FPS and VBench Score that split 5s videos into 3 chunks. Point color indicates the average temporal starting position of chunks, from earlier (blue) to later (yellow).

configurations that can be tailored to diverse compute budgets and videos of different lengths, and offers a more favorable trade-off between quality and efficiency.

We perform a brute-force search to identify the optimal chunk configuration for a 5-second video under a fixed constraint of three chunks. Concretely, we enumerate all valid partitions of the 21 latent frames into three chunks, excluding cases that a chunk only contain one frame. The results are shown in Figure 4, from which we draw the following key observations.

- Uniform chunking is not optimal. Evenly partitioning frames can lead to markedly inferior performance compared to asymmetric alternatives.

- Chunk layout matters beyond exposure bias. Under the same number of exposure rounds, different chunk configurations exhibit large performance disparities.

- Performance favors temporally front-loaded chunking. Allocating larger chunks to early frames and smaller chunks to later frames yields the strongest results, occasionally surpassing bidirectional inference.

### 4.2. Autoregressive Any-timestep, Any-order Editing

Under Flex-Forcing, we unlock two new forms of editing that are difficult under conventional autoregressive paradigm. Specifically, we explore two directions: autoregressive any-timestep and any-order editing.

**Autoregressive any-timestep editing.** To maintain global coherence, we adopt a structure-preserving editing strategy that restricts editing to low-level refinement timesteps while keeping high-level planning timesteps unchanged. This

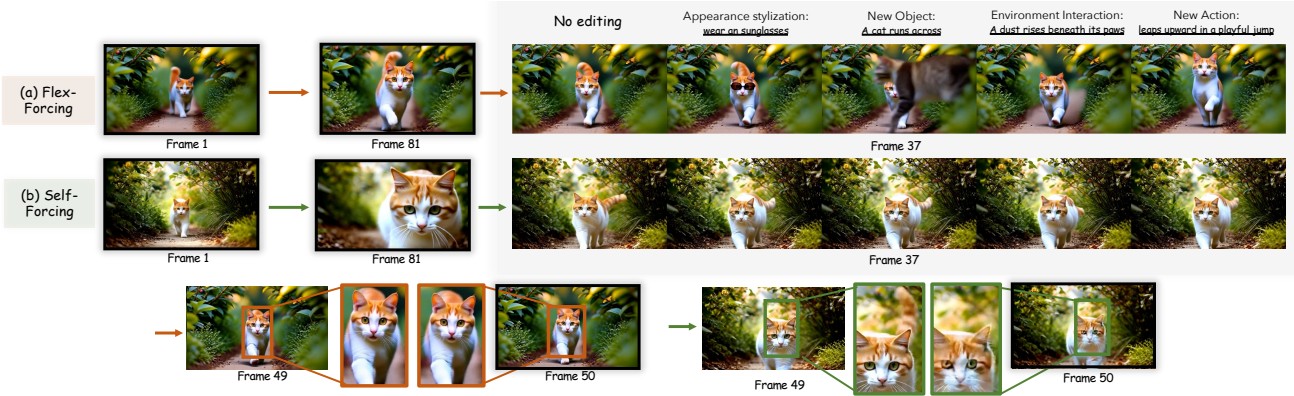

*Figure 5.* Autoregressive any-order generation under the Flex-Forcing paradigm. We first generate the full sequence of 81 frames and then re-edit an arbitrary temporal segment (frames 26 – 49). Our method achieves a high success rate for editing while preserving significantly better consistency with the unedited subsequent chunks (see the transition between Frame 49 and Frame 50).

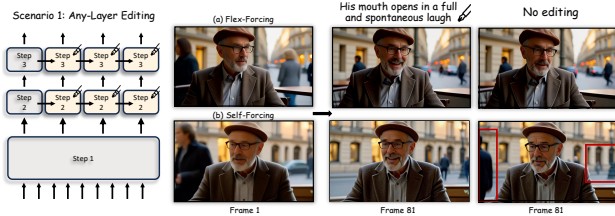

*Figure 6.* Autoregressive any-timestep editing. We modify the conditioning starting from the 6th frame during the last three of four denoising steps. For a fair comparison, the baseline applies the same conditioning updates at the identical frames and timesteps.

separation explicitly decouples global structure formation from local detail modification, resulting in substantially improved consistency over baseline autoregressive editing methods (Figure 6). Under the Self-Forcing paradigm, the globally coupled generation dynamics cause even small local edits to propagate across frames, often leading to large deviations in the final outputs.

**Autoregressive any-order editing.** Leveraging the ability to attend to both past and future tokens, our model enables any-order editing while still operating within an autoregressive generation framework. Editing a target chunk can thus be performed independently of its temporal order. Formally, editing at timestep $t$ and chunk $k$ is defined as:

$$x_{t-1}^{\mathcal{F}_{t,k}} \sim q_\theta \left( x_{t-1}^{\mathcal{F}_{t,k}} \mid x_0^{<\mathcal{F}_{t,k}}, x_0^{>\mathcal{F}_{t,k}}, x_t^{\mathcal{F}_{t,k}} \right) \quad (7)$$

where $< \mathcal{F}_{t,k}$ and $> \mathcal{F}_{t,k}$ represents clean tokens belonging to chunks before and after $\mathcal{F}_{t,k}$. Conditioning on both sides allows arbitrary temporal segments to be re-edited after full-video generation without re-generating the entire sequence, relaxing the strict causal constraint of standard autoregressive models (Figure 5).

| Model | #Params | FPS | NFE | Total | Quality | Semantic |
|---|---|---|---|---|---|---|
| *Pre-trained Autoregressive / Diffusion Video Generation Models* | | | | | | |
| SkyReels-V2 (Chen et al., 2025a) | 1.3B | 2.2 | 30×2 | 82.67 | 84.70 | 74.53 |
| MAGI-1 (Teng et al., 2025) | 4.5B | 0.5 | 64×3 | 79.18 | 82.04 | 67.74 |
| NOVA (Deng et al., 2024) | 0.6B | 0.9 | 25×2 | 80.12 | 80.39 | 79.05 |
| Pyramid Flow (Jin et al., 2024) | 2B | 0.9 | 20×2 | 81.72 | 84.74 | 69.62 |
| Wan2.1 (Wan et al., 2025) | 1.3B | 1.3 | 50×2 | 84.26 | 85.30 | 80.09 |
| *Autoregressive Model Distilled from Diffusion Models* | | | | | | |
| CausVid (Yin et al., 2025) | 1.3B | 24.9 | 5 | 81.20 | 84.05 | 69.80 |
| Self Forcing (Chunk-wise) | 1.3B | 24.9 | 5 | 84.31 | 85.07 | 81.28 |
| Self Forcing (Chunk-wise)* | 1.3B | 24.9 | 5 | 83.89 | 84.45 | **81.63** |
| Self Forcing (Frame-wise) | 1.3B | 24.9 | 5 | 84.26 | 85.25 | 80.30 |
| Self-Forcing++ (Cui et al., 2025) | 1.3B | 24.9 | 5 | 83.11 | 83.79 | 80.37 |
| Ours (15-3-3) - Best Performance | 1.3B | 25.8 | 5 | **85.07** | **86.33** | 80.02 |
| Ours - (7-7-7) - Fastest | 1.3B | 29.4 | 5 | 84.63 | 85.89 | 79.59 |
| Ours (3-3-3-3-3-3) | 1.3B | 24.9 | 5 | 84.03 | 84.91 | 79.95 |
| Ours (12-6-3) - Best Performance | 1.3B | 45.6 | 3 | 84.29 | 85.33 | 80.12 |
| Ours - (7-7-7) - Fastest | 1.3B | 48.9 | 3 | 83.92 | 84.96 | 79.78 |
| Ours (3-3-3-3-3-3) | 1.3B | 41.5 | 3 | 83.91 | 84.85 | 80.16 |

*Table 2.* Comparisons of performance for 5s videos. *: We sample videos from the official checkpoint and test its performance. Here, the NFE of the causal distillation method contains $N$ steps for denoising and 1 step for caching.

# 5. Experiments

**Benchmark and baselines** We build our baseline on Wan2.1-T2V-1.3B (Wan et al., 2025), which generates 5-second videos at 832×432 resolution. We evaluate inference efficiency and generation quality. Efficiency is measured in FPS for 81-frame generation on GB200. Quality is assessed using VBench (Huang et al., 2024) for 5-second videos and VBench-Long (Huang et al., 2025b) for 30-second videos. For fair comparison, we use the same prompts as Self-Forcing and fix random seeds across all experiments. We generate 5 samples per prompt for 5-second videos and one sample per prompt for 30-second videos.

We consider two groups of baselines. For the 5-second set-

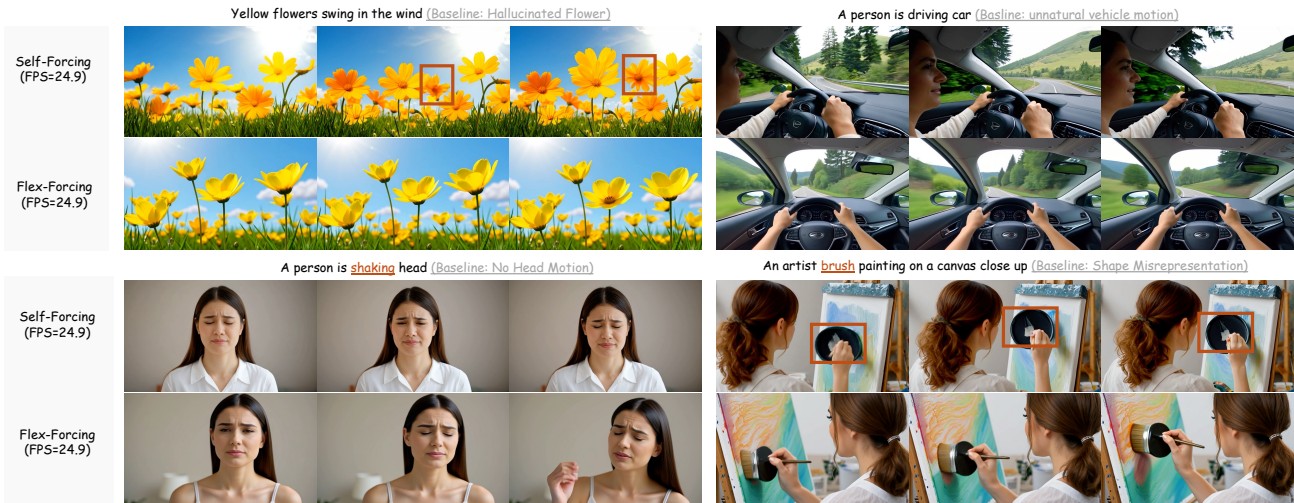

*Figure 7.* Visual comparison between Self-Forcing and Flex-Forcing in generating 5s videos. We show three frames at 0 s, 2.5 s, and 5 s. Flex-Forcing produces more coherent long-range motion and better alignment with the given instruction.

| Model | Steps | Evaluation Scores | | |
| --- | --- | --- | --- | --- |
| | | Total | Quality | Semantic |
| Wan2.1 | 50×2 | 84.26 | 85.30 | 80.09 |
| DOLLAR (Ding et al., 2025) | 4 | 82.57 | 83.83 | 77.51 |
| T2V-Turbo-v2 (Li et al., 2024) | 4 | 82.34 | 83.93 | 75.97 |
| rCM (Zheng et al., 2025) | 4 | 84.43 | 85.38 | 80.63 |
| DMD-v (Nie et al., 2026) | 4 | 84.60 | 86.03 | 79.87 |
| Flex-Forcing | 4 | 85.13 | 86.50 | 79.65 |
| TMD (Nie et al., 2026) | 2.33 | 84.68 | 85.71 | 80.55 |
| APT (Lin et al., 2025) | 2 | 81.85 | 84.39 | 71.70 |
| rCM (Zheng et al., 2025) | 2 | 84.09 | 84.90 | 80.86 |
| DMD-v (Nie et al., 2026) | 2 | 84.39 | 85.65 | 79.32 |
| Flex-Forcing | 2 | 84.20 | 85.27 | 79.92 |

*Table 3.* Comparisons with few-step distilled diffusion models.

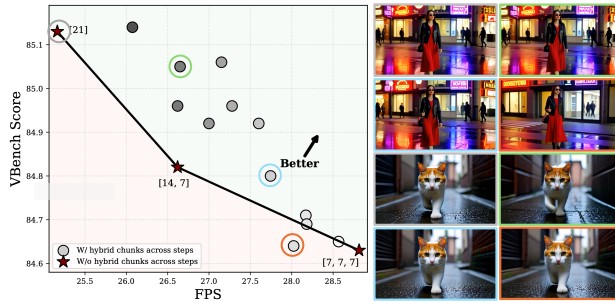

*Figure 8.* Results for hybrid chunking over denoising time. The points with lighter color represents those with those have more fine-grained splits.

is 21 latent frames.

ting, we compare against Self-Forcing (Huang et al., 2025a) as well as recently proposed improvements built upon it (Cui et al., 2025). Moreover, since our model also supports bidirectional diffusion inference, we additionally compare with few-step distillation methods for video diffusion models. For long-video evaluation, we compare our method with approaches that are train-short-evaluate-long (Huang et al., 2025a; Yesiltepe et al., 2025).

**Implementations**  We follow Self-Forcing to use Wan2.1-T2V-1.3B as our base model and Wan2.1-T2V-14B as the teacher model to train Flex-Forcing. Training uses extended prompts from VidProM (Wang & Yang, 2024) for 600 iterations with a batch size of 64. We randomly pick the chunk size in training from 2 to 10. For the K-projection, we use a learning rate of 2e-6. For the 2-step model, we set the denoising steps set to $[1000, 500]$. The sink size for the long video is set to 3 latent frames, and the total attention window

**Performance on 5s videos**  Results on 5-second videos are summarized in Tables 2 and 3. For Flex-Forcing, we report three representative configurations: the best-performing setting, the speed-optimized setting, and the configuration matched to the baseline. Flex-Forcing consistently outperforms the baselines in both inference efficiency and video quality. We further compare Flex-Forcing with few-step distilled diffusion methods. Even in this regime, Flex-Forcing achieves comparable or superior video quality, indicating that the proposed framework preserves strong performance under fully bidirectional operation. Qualitative case studies are presented in Figure 7. The primary improvements arise in temporal dynamics: long-horizon planning enables coherent long-range motion, mitigates error accumulation, and yields smoother and more temporally consistent generation.

**Performance of hybrid chunking over timesteps**  We further analyze the effect of hybrid chunking on denoising timesteps through an ablation study. Specifically, we consider three chunking configurations, $[21]$, $[14, 7]$ and

| | Subject C. | Human Action | Aesthetic Quality | Temporal Style | Overall C. | Background C. | Appearance Style | Scene | Quality Score | Semantic Score |
|---|---|---|---|---|---|---|---|---|---|---|
| LongLive* | 98.36 | 97.15 | 63.77 | 24.15 | 26.49 | 96.25 | 20.52 | 58.62 | 83.21 | 81.17 |
| Self-Forcing | 97.76 | 96.86 | 61.72 | 23.56 | 25.68 | 95.00 | 21.15 | 51.05 | 82.67 | 76.77 |
| Infinity-RoPE | 97.68 | 96.99 | 62.33 | 23.74 | 26.26 | 96.00 | 20.91 | 53.47 | 83.77 | 79.11 |
| Ours | 98.00 (0.40↑) | 97.07 (0.08↑) | 63.31 (0.98↑) | 24.26 (0.48↑) | 25.92 (0.34↓) | 95.50 (0.50↓) | 20.32 (0.59↓) | 52.20 (1.27↓) | 85.30 | 78.86 |
| | Imaging Quality | Color | Object Class | Flickering | Motion Smoothness | Dynamic Degree | Multiple Objects | Spatial | Total Score | FPS |
| LongLive* | 68.35 | 90.07 | 94.08 | 99.36 | 98.81 | 38.89 | 85.86 | 82.06 | 82.80 | 25.07 |
| Self-Forcing | 67.70 | 71.55 | 92.83 | 99.10 | 98.67 | 41.93 | 83.65 | 76.55 | 81.49 | 19.10 |
| Infinity-RoPE | 69.30 | 82.52 | 91.90 | 99.28 | 98.72 | 50.26 | 83.55 | 82.48 | 82.84 | 19.10 |
| Ours | 66.24 (3.06↓) | 84.92 (2.40↑) | 93.62 (1.72↑) | 99.70 (0.42↑) | 98.70 (0.02↓) | 71.27 (21.01↑) | 85.15 (1.60↑) | 78.11 (4.37↓) | 84.01 | 24.96 |

*Table 4.* VBench-Long comparison on 30-second videos. *Except for LongLive, all other methods are trained only on short videos.

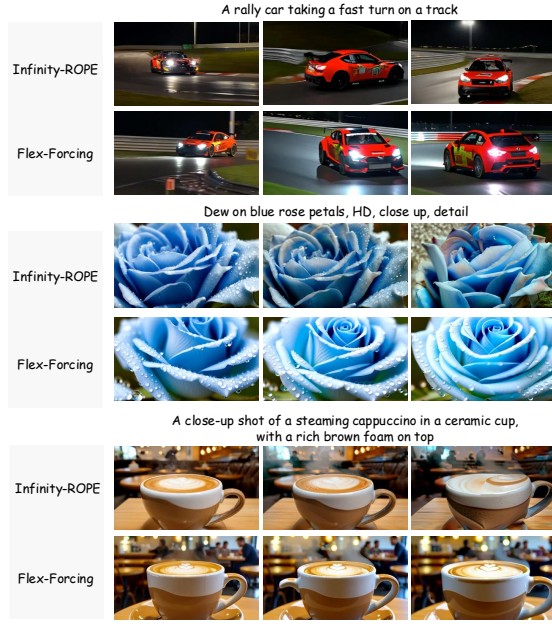

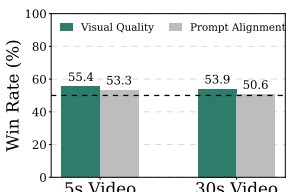 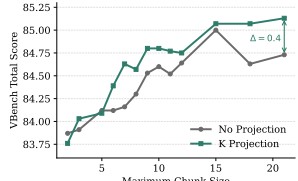

*Figure 10.* (Left) User study: We compare Flex-Forcing with Self-Forcing under 5s and 30s settings. (Right) Ablation of K-projection and its impact on different chunk configurations.

*Figure 9.* Comparisons between Flex-Forcing and Infinity-RoPE. Our method produces long video with improved semantic consistency, better aesthetic quality and mothion smoothness.

[7, 7, 7], where each configuration specifies the chunk sizes assigned to the denoising steps. We evaluate all settings under identical conditions, and the quantitative results are summarized in Figure 8. Overall, hybrid chunking consistently achieves a more favorable trade-off between efficiency and quality compared to applying chunking only along temporal frame, effectively yielding a better Pareto frontier. We conduct qualitative analysis and observe that even the weakest-performing configuration, where the first denoising step uses a chunk size of 21 and the remaining stages use a chunk size of 7, still significantly outperforms Self-Forcing (84.3), while producing video content and fine-grained details similar to that of fully bidirectional model.

**Performance on 30s videos**  Table 4 shows the evaluation results of our method under the 30s setting. We build the inference of our method upon Infinity-RoPE (Yesiltepe et al., 2025). We observe that our method outperforms Infinity-RoPE across the majority of evaluation metrics,

while also achieving faster inference speed. Compared to Infinity RoPE, our improvements primarily stem from enhanced video quality rather than semantic alignment, as our method does not explicitly optimize for alignment objectives. Notably, the most significant gains are observed in the dynamic degree, where our approach demonstrates a clear advantage. We further provide qualitative comparisons in the appendix to illustrate these improvements, showing substantially richer and more expressive motion dynamics, where the baseline tends to generate repetitive motions, resulting in reduced dynamic diversity.

**User preference study**  As shown in Figure 10, our method are preferred than self-forcing both on the 5s and 30s setting. The advantage is more significant for the dimension of visual quality.

**Ablation study: impact of K-projection**  We evaluate the impact of K-projection by comparing models trained with and without it under various inference configurations (Figure 10). Incorporating K-projection consistently improves performance across all settings, with the largest gains observed near the fully bidirectional regime. Without K-projection, performance degrades as chunk size increases; with K-projection, performance remains stable and improves smoothly. This demonstrates that K-projection stabilizes training when approaching bidirectional supervision and mitigates performance drops at large chunk sizes.

## 6. Conclusion

In summary, Flex-Forcing reframes bidirectional and autoregressive video generation not as competing paradigms, but a controllable inference dimension that can be smoothly adjusted according to requirements. By introducing the flexible chunking along the temporal and denoising axes, and bridging the different level of noise in causal and non-causal contexts, the model can adapt its causal structure at test time, enabling better globally coherent generation, efficient and better long-video synthesis and better trade-offs between quality and speed.

## Limitations

Our method still has several limitations. Although it relaxes the strict left-to-right training constraint, the training–inference mismatch is not fully resolved and can still accumulate errors in long video generation. In addition, the effectiveness of the method appears to depend heavily on capabilities inherited from pre-training, especially bidirectional encoding priors, which may limit its transferability to models where such priors are weak or absent. More broadly, as with other generative video models, the proposed method may be misused in downstream applications if deployed without appropriate safeguards.

## Impact Statement

Flex-Forcing unifies bidirectional and autoregressive paradigms into a single model, reducing the computational cost and environmental footprint of maintaining separate specialized architectures. By allowing dynamic trade-offs between speed and quality at runtime, it democratizes high-fidelity, long-form video generation for resource-constrained environments. Additionally, the order-agnostic capabilities provide precise, localized editing tools for creative and industrial sectors, promoting more adaptive, resource-aware generative systems in media, simulation, and embodied AI.

## Acknowledgement

This research is in part supported by the Ministry of Education, Singapore, under the Academic Research Fund Tier 1 (FY2026, WBS: A-8004345-00-00).

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

# A. Qualitative results on flexible chunks on video frames.

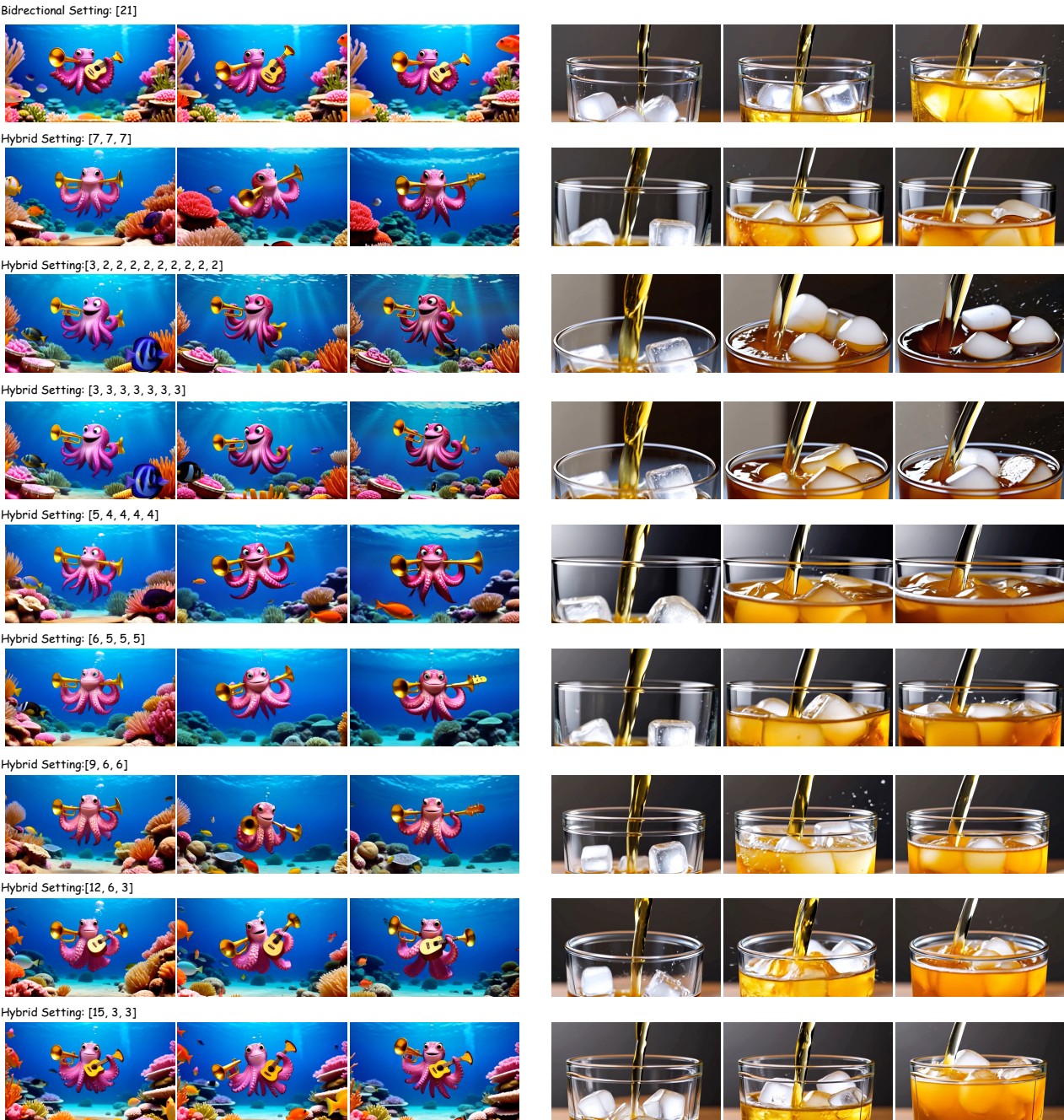

*Figure 11.* **Prompt (left):** A vibrant and lively underwater scene featuring an octopus playing multiple musical instruments simultaneously in a colorful band. The octopus has a playful and joyful expression, its tentacles deftly manipulating a trumpet, a drum, and a guitar. Its body is adorned with iridescent patterns, and it appears to be having fun. The background showcases a diverse array of marine life, including colorful fish and coral reefs, with a gentle underwater current flowing. The scene is captured in a dynamic angle, emphasizing the octopus's movements and the instruments it plays. The water has a soft, shimmering quality, enhancing the underwater atmosphere. A mid-shot with a dynamic camera angle. **Prompt (right):** A close-up shot of a drink being poured over ice, showcasing the detailed flow of liquid interacting with the ice cubes. The drink cascades down, creating ripples and splashes on the surface of the ice, which glistens under the soft lighting. The glass holds a clear, amber-colored liquid, and the ice cubes sparkle with tiny droplets of condensation. The background is blurred, highlighting the dynamic interaction between the drink and the ice. The photo has a crisp, natural lighting style, emphasizing the fluid motion and the sparkling ice. A close-up from a slightly downward angle.

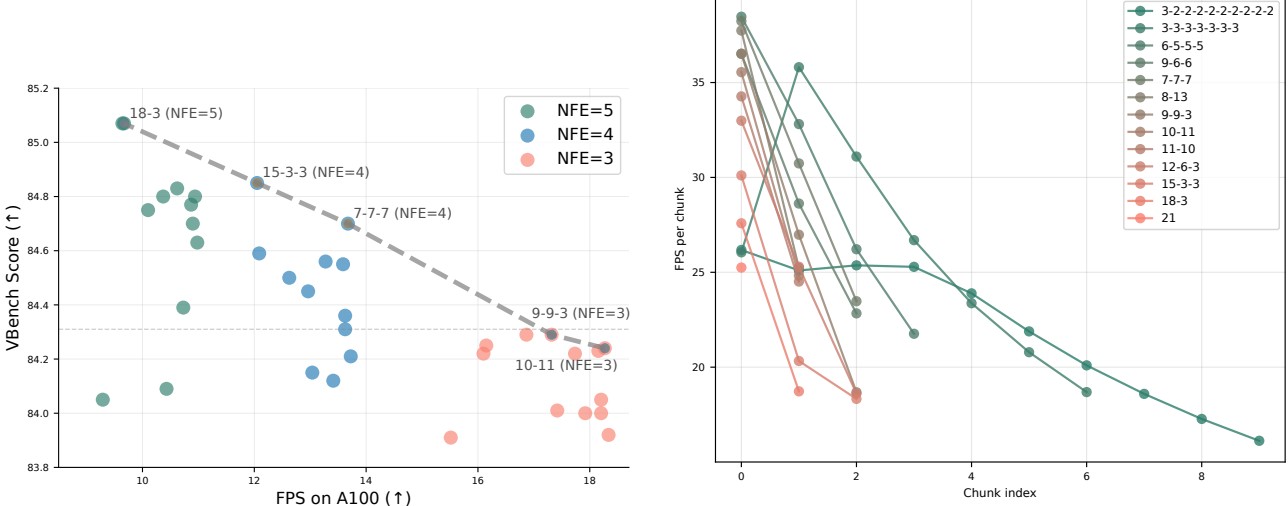

*Figure 12.* (Left) The VBench score and FPS on a A100 GPU. We adopt the same configurations as on GB200. (Right) We plot the FPS in each chunk if we have different chunk configurations.

In the experiment that shown in Figure 1, we use the following 14 configurations: [21], [18,3], [15,3,3], [12,6,3], [11,10], [10,11], [9,9,3], [8,13], [8,8,5], [7,7,7], [9,6,6], [6,5,5,5], [5,4,4,4], [3,3,3,3,3,3,3], [3,2,2,2,2,2,2,2,2,2]. We here show some results when we set the chunking configurations to those difference choices in Figure.11. In these experiments, we use the same model with an identical random seed and vary only the chunking configuration. The results reveal a clear and consistent trend: as the average chunk size increases, the visual quality of the generated videos improves steadily and progressively approaches that of full bidirectional attention. In contrast, when the chunk size is small, we observe pronounced exposure bias, indicating that exposure bias still remains a significant issue under fine-grained chunking.

**Speed Analysis** We further provide a detailed speed analysis. All experiments are conducted on a single A100 GPU, and the results are reported in Figure 12. We then show the results if we calculate the FPS for each block. While smaller chunk sizes achieve higher per-chunk FPS, they require a larger number of rollout rounds, resulting in a lower overall throughput. In contrast, although each chunk operates at a relatively lower FPS, the total number of rollout rounds is substantially reduced compared to Self-Forcing, leading to higher end-to-end generation speed.

## B. Qualitative results on flexible chunks on denoising timesteps.

Figure 13 illustrates the processing order when applying flexible chunking over denoising timesteps. The 0-th block is bidirectional and thus does not depend on any prior KV cache, allowing immediate execution. In contrast, although Block 4 has access to intermediate representations from Block 1, it cannot be computed until Block 3 finishes, since the required KV caches from earlier frames are unavailable. We further study the impact of flexible chunking across denoising steps under multiple configurations. As shown in Figure 14, we evaluate three chunking schemes—[21], [14, 7], and [7, 7, 7]. All configurations yield results highly similar to the unchunked bidirectional setting, with differences mainly in local visual details. As chunking becomes more fine-grained, deviations from the bidirectional baseline become more evident, especially in later frames, suggesting reduced temporal consistency toward the end of the video.

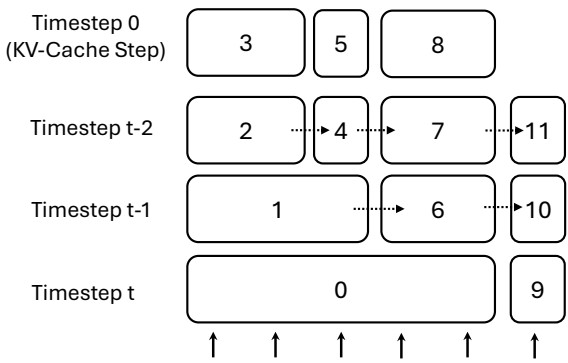

*Figure 13.* Illustration of flexible chunking across timesteps. The number on the block is the order they would be executed.

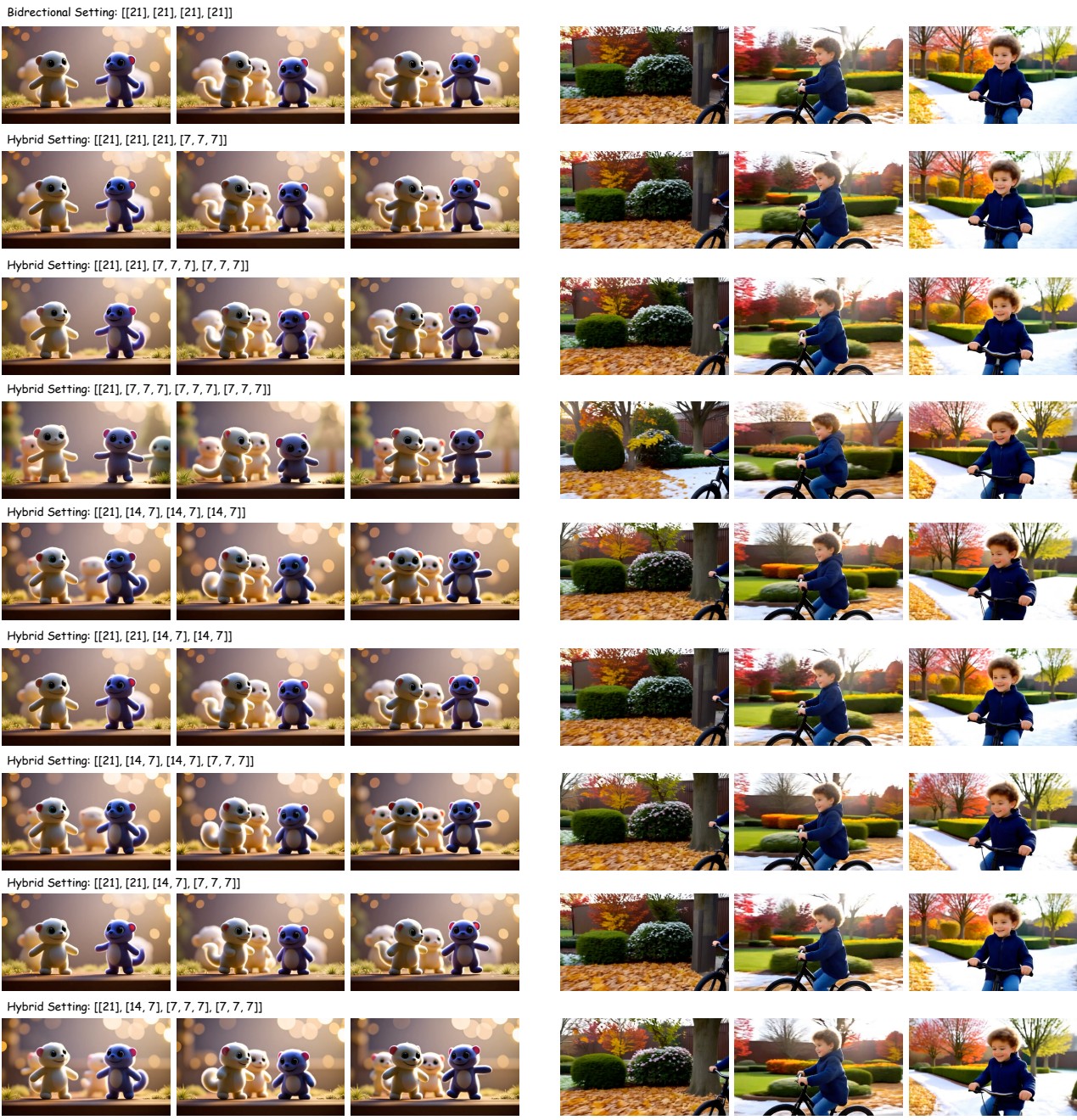

*Figure 14.* **Prompt (left):** A miniature 3D render in an octane engine style depicting adorable wool and felt monsters dancing together in a dreamy, bokeh-filled setting. These soft, cuddly creatures, with big expressive eyes and fluffy bodies, are illuminated by gentle, diffused lighting that casts a warm, ethereal glow. The background features a soft, hazy backdrop with a dreamy bokeh effect, adding a cinematic quality to the scene. The monsters are shown from various angles, capturing their playful movements and expressions, creating a charming and enchanting atmosphere. A medium shot with a dynamic camera angle, highlighting the natural and joyful dance of these woolen monsters. **Prompt (right):** A dynamic photograph capturing a little boy riding his bike through a garden that transitions through the changing seasons—fall leaves crunch underfoot, winter snow blankets the ground, spring flowers bloom, and summer sunshine sparkles through the foliage. The boy, with curly brown hair and a joyful smile, pedals energetically, his arms outstretched in excitement. The garden backdrop features trees with branches adorned in each season's distinctive foliage. A series of shots taken from various angles, starting with a wide shot of the boy entering the garden in spring, transitioning to a mid-shot of him biking through the colorful autumn leaves, then a close-up of him riding through a snowy path, and finally a wide-angle view of him enjoying the warm summer sun. The photo has a natural, documentary style, emphasizing the boy's natural movements and the vibrant colors of the changing seasons.

**Self-Forcing**                                           **Flex-Forcing**

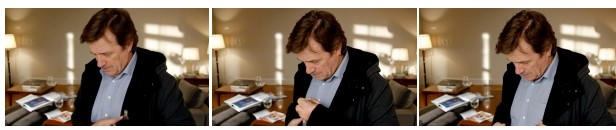 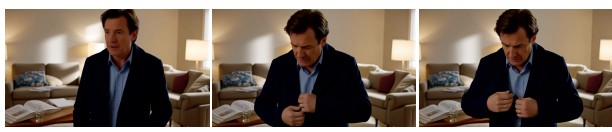

**Prompt:** A candid moment captured in a documentary-style photo of a middle-aged man looking bewildered and slightly frustrated as he searches his pockets and coat for his missing keys. He stands in a cluttered living room with books and magazines scattered on a coffee table, and a half-empty glass on the floor. His face is filled with worry, and his fingers run through his tousled brown hair. The background shows a mix of shadows and bright spots from nearby lamps, creating a warm yet anxious atmosphere. A close-up shot from a low angle, emphasizing his expression.

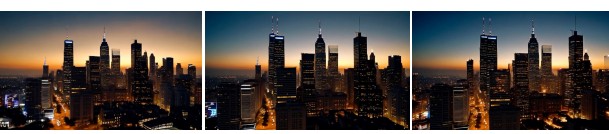 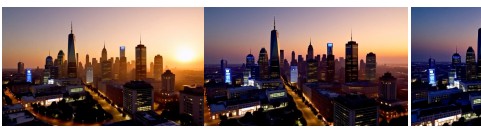 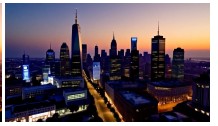

**Prompt:** A cinematic pan right over a bustling city skyline at dusk, capturing the transition from day to night. The buildings begin to twinkle with lights as the sun sets below the horizon, casting a warm golden glow over the scene. The camera gradually widens, revealing the intricate details of skyscrapers, illuminated billboards, and the busy streets below. A soft haze in the air adds depth and a sense of mystery to the urban landscape. The overall style is reminiscent of a Hollywood evening promotional poster, with a blend of realistic and slightly exaggerated architectural details. A sweeping medium shot with a dynamic camera movement.

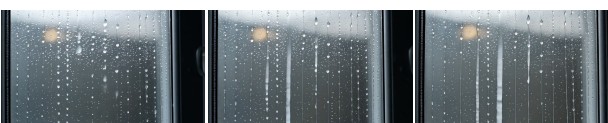 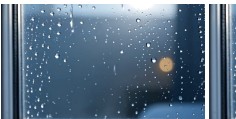 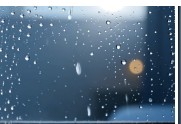 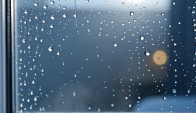

**Prompt:** A close-up of steam condensing on a cold glass windowpane, with tiny droplets merging and sliding away as they gather. The glass is clear, showing the condensation forming into small beads that roll down the surface. The background is dimly lit, with only the soft glow of interior lights visible, creating a misty and ethereal atmosphere. The camera angle is slightly tilted downward, capturing the droplets' movement and the subtle play of light on the glass.

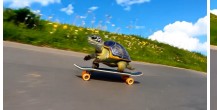 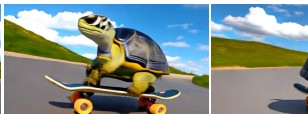 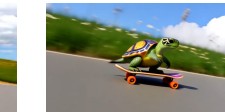 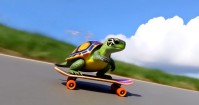 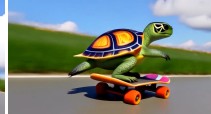

**Prompt:** A dynamic action shot of a turtle wearing a sleek racing suit, riding a colorful skateboard down a steep hill. The turtle has a determined expression, its small legs pumping vigorously as it skates with speed and agility. The skateboard wheels spin rapidly, leaving a slight blur in the background. The hillside is lined with tall grass and wildflowers, and the sky is a bright blue with fluffy clouds. The scene captures the turtle's momentum and excitement as it races down the hill, with a slight tilt to the camera angle to enhance the sense of speed and movement.

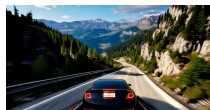 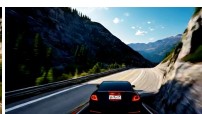 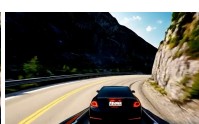 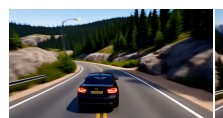 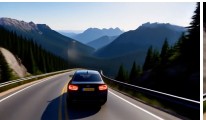 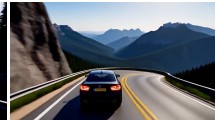

**Prompt:** A dynamic tracking shot in the style of a thrilling action movie, capturing a car navigating a winding mountain road. The camera follows the car closely, showcasing the rugged terrain and scenic views. As the car twists and turns, the landscape changes dramatically, revealing lush green forests, steep cliffs, and distant peaks. The road winds through valleys and over rocky outcrops, creating a sense of adventure and excitement. The car's headlights illuminate the path ahead, casting shadows on the rugged landscape. The overall scene is rendered in a high-definition, cinematic style, emphasizing the movement and the breathtaking vistas.

*Figure 15.* Visual Comparisons between Flex-Forcing and Self-Forcing.

## C. Case Study

we provide further comparisons between Self-Forcing and Flex-Forcing to complement the main results (See Figure 15). We also compare Flex-Forcing under different numbers of function evaluations with NFE=5 and NFE=3 (See Figure 16), to analyze the impact of denoising steps on generation quality and efficiency.

## D. Impact Statement

This work advances the foundations of video generation by demonstrating that bidirectional diffusion and autoregressive generation, traditionally treated as mutually exclusive paradigms, can be unified within a single, flexible model. By enabling dynamic trade-offs between generation quality, efficiency, and controllability at inference time, the proposed framework reduces the need to train and maintain multiple specialized models, thereby lowering computational cost and environmental footprint. Beyond improving scalability for long-form video generation, this flexibility unlocks new applications such as localized, order-agnostic video editing and efficient refinement under constrained budgets. We anticipate that these capabilities will benefit a wide range of downstream domains, including content creation, simulation, and embodied AI, while encouraging future research toward adaptive, resource-aware generative systems that better align model capabilities with real-world constraints.

A baby is learning to walk with his mother

A close-up of paint being mixed, showing the detailed interaction of colors and textures.

*Figure 16.* Visual Comparisons between Flex-Forcing with NFE=5 and NFE=3.

