# OpenReview forum: "Flex-Forcing: Towards a Unified Autoregressive and Bidirectional Video Diffusion Model"
_ICML.cc/2026/Conference — ICML 2026 spotlight_

### Official Review · Reviewer_8MRG · 2026-02-23

**Soundness:** 3
**Presentation:** 3
**Significance:** 3
**Originality:** 3
**Overall Recommendation:** 5
**Confidence:** 4

**Summary:**

This paper proposes Flex-Forcing, a post-training framework that lets one video diffusion model run in bidirectional, autoregressive, or hybrid modes at inference time. They demonstrate improved quality–speed trade-offs vs Self-Forcing on Wan2.1-T2V-1.3B.

**Compliance With Llm Reviewing Policy:**

Affirmed.

**Key Questions For Authors:**

● 1) Generalization: How does Flex-Forcing perform on other backbones (not Wan2.1) ?

● 2) Chunk search practicality: Can you provide a simple rule-of-thumb or learned policy for picking chunk configs, instead of brute-force search?

● 3) Long-video stability: Beyond VBench-Long averages, can you report failure rates / drift metrics vs video length?

**Limitations:**

yes

**Strengths And Weaknesses:**

● Soundness: (pros) The core framing is technically reasonable: chunking induces a continuum between fully bidirectional and fully causal inference, and K-Projection directly targets a mismatch. (cons) Some parts feel a bit “engineering”, for example, the method relies on a search over chunk configurations on specific Wan models. User study shows that the advantage over self-forcing is small.

● Presentation: (pros) The paper is mostly easy to follow. (cons) Reproducibility would benefit from a single “default config” table that spells out training/inference schedules, window/sink sizes, chunk sampling ranges, and exactly what’s fixed vs searched. Some key setup definitions are not clear, for example the meaning of (15-3-3) and (3-3-3-3-3-3-3).

● Significance: The problem studied in this paper a meaningful for real applications.

● Originality: The components aren’t entirely new (chunked causal generation, distillation, KV caching), but the two-axis chunking + noise-level alignment (K-Projection) is a pretty novel combination.

---

> ### Author Rebuttal · Authors · 2026-03-31
>
> We sincerely thank the reviewer for the constructive comments and for the time and effort devoted to reviewing our manuscript.
>
> > W1: Some parts feel a bit “engineering”, for example, the search over chunk configurations. Q2: Can you provide a simple rule-of-thumb or learned policy for picking chunk configs?
>
> Thank you for your valuable comments. We agree that the chunking strategy is somewhat engineering in its current form. Our main goal in including this experiment is to demonstrate the potential benefit of flexible chunking: as shown in Figure 4, 73\% of the tested configurations outperform the Self-Forcing baseline, and can also surpass the bidirectional WAN baseline (84.26 v.s. 85.13).
>
> We have not yet explored a learned chunking policy. One reason is that the empirical pattern of effective chunking strategies already appears relatively clear: for a fixed number of chunks, allocating more frames to earlier chunks and fewer frames to later chunks tends to work better. For example, under a three-chunk setting, the configuration [9, 6, 6] outperforms [7, 7, 7] by around 0.5 VBench points. And we also adopt this rule for SANA-Video, showing that it can also surpass the baseline.
>
> We agree this is an interesting direction, and learning or automatically selecting chunking strategies would be a valuable extension for future work.
>
> > W2: Reproducibility would benefit from a single “default config” table. Some key setup definitions are not clear.
>
> **Response**: Thanks for pointing this out. We would add a configuration table in the appendix to show the training and inference configuration to make the experiments more reproducible. Notations such as (15-3-3) and (3-3-3-3-3-3-3) denote the number of frames contained in each chunk. For example, (15-3-3) indicates that the video is divided into three chunks containing 15, 3, and 3 frames, respectively.
>
> > Q1: Generalization: How does Flex-Forcing perform on other backbones (not Wan2.1) ?
>
> Thanks for your great suggestion. We provide results on SANA-Video as an additional backbone for testing. This backbone differs substantially from Wan2.1 that it adopts linear attention as its core attention mechanism and is pretrained with a different training recipe. Flex-Forcing remains effective on SANA-Video, and we provide two sets of results below.
>
> * Comparison with Self-Forcing on SANA-Video. Our method exceeds the LongSANA trained with self-forcing.
>
> | | Quality|Semantic|Total|
> | - | - | - | - |
> |LongSANA Self-forcing (Official checkpoint) |82.29|77.17|81.27|
> |SANA-Video Flex-Forcing |82.33|79.30|81.72|
>
> Training and evaluation configuration: We evaluate SANA-Video Self-Forcing using the officially released LongSANA-Video-2B-Self-Forcing checkpoints and the official inference configuration. We do not perform the subsequent LongSANA training stage after self-forcing, so some performance gap is expected compared with models that undergo the full LongSANA training pipeline, as reported in the paper.
>
> * Flexibility under different chunking strategies:
>
> We further verify that Flex-Forcing preserves its key advantage of flexible chunking at inference time. We evaluate the trained model under multiple chunking strategies. Most of the results surpass the performance of official self-forcing checkpoint of LongSANA (Total score = 81.27)
>
> |Chunk Strategy |Quality|Semantic|Total|
> |-|-|-|-|
> |12-9 | 82.27 | 78.66 | 81.55|
> |13-8 | 82.33 | 78.94 | 81.66|
> |7-7-7 | 82.00 | 78.91 | 81.38|
> |9-6-6 | 82.14 | 79.01 | 81.52|
> |6-5-5-5 | 81.94| 79.11| 81.37|
> |3-3-3-3-3-3-3 | 80.23| 79.58| 80.10|
>
> > Q3: Long-video stability: Beyond VBench-Long averages, can you report failure rates / drift metrics vs video length?
>
> Thanks for your great suggestion. We here use one drift metric to quantify drifting from a statistical perspective. Specifically, we measure color distribution drift using histogram distance in Lab space. Each frame is converted to the Lab color space, and a 3D color histogram is computed. It is then compared with the first frame using the chi-square distance. A smaller value indicates less shift in the color distribution:
>
> - Results on Wan2.1:
> || 5s | 10s | 20s |  30s |
> | - | - |   - |  - |  - |
> | Wan2.1 - Self-Forcing | 0.051 | 0.114 | 0.207 | 0.262 |
> | Wan2.1 - Flex-Forcing | 0.039 | 0.078 | 0.149 | 0.199 |
>
> - Results on Sana-Video:
> || 5s | 10s | 20s |  30s |
> | - | - |   - |  - |  - |
> | Sana - Self-Forcing | 0.068 | 0.175 | 0.314 | 0.404 |
> | Sana - Flex-Forcing | 0.043 | 0.100 | 0.211 | 0.303 |

---

> > ### Author Rebuttal · Reviewer_8MRG · 2026-04-01
> >
> > most of my concerns are resolved

---

### Official Review · Reviewer_4fb8 · 2026-03-04

**Soundness:** 3
**Presentation:** 3
**Significance:** 2
**Originality:** 3
**Overall Recommendation:** 4
**Confidence:** 4

**Summary:**

This paper proposes Flex-Forcing, a framework that unifies autoregressive and bidirectional video diffusion generation within a single model. The key idea is a flexible chunking strategy that partitions video frames across both the temporal dimension and diffusion timesteps. During inference, chunks are generated autoregressively while frames within a chunk are modeled bidirectionally, enabling different trade-offs between global coherence and inference efficiency. The method also introduces a noise-alignment mechanism (K-projection) to reconcile representations from tokens at different noise levels. Experiments on video generation benchmarks demonstrate improvements in quality and efficiency compared to Self-Forcing and other baselines.

**Compliance With Llm Reviewing Policy:**

Affirmed.

**Final Justification:**

The paper is technically solid,  and the idea of flexibly combining autoregressive and bidirectional generation is simple but useful . I still see limitations in time-to-first-frame and evaluation breadth, especially beyond a single model scale. The rebuttal adds helpful experiments and clarifications, but only partially addresses these concerns and does not change my overall view. Overall, I consider this a good but not fully mature contribution, and maintain a weak accept.

**Key Questions For Authors:**

1. If time-to-first-frame or streaming constraints are not a concern, it is unclear whether the proposed hybrid approach provides a clear advantage over simply using a fully bidirectional diffusion model. In that case, a bidirectional model might achieve the best quality–efficiency trade-off?

**Limitations:**

The author did not include limitation of their method?

**Strengths And Weaknesses:**

**Strengths**

1. Soundness: The method shows consistent improvements over Self-Forcing across several configurations. The reported results suggest a better Pareto frontier between generation quality and throughput.

2. Presentation: The paper is well organized and easy to follow. The motivation for combining autoregressive and bidirectional paradigms is well articulated, and the experimental section is fairly comprehensive.

3. Originality: The paper explores a hybrid generation paradigm that bridges autoregressive and bidirectional video diffusion. The flexible chunking perspective is conceptually simple yet potentially useful for controlling the quality–efficiency trade-off.

4. Significance: see weekness.

**Weakness**
1. Significance: The proposed approach introduce higher latency in time-to-first-frame compared to purely autoregressive generation, since frames within a chunk are processed jointly before output. This could limit its applicability in streaming or interactive scenarios. Please correct me if I am wrong on this.

2. Soundness: All experiments are conducted on Wan 1.3B. It is unclear whether the approach continues to provide benefits at larger model scales or with different architectures. The evaluation primarily relies on VBench metrics. It would be helpful to include human evaluations to better assess perceptual quality and motion coherence, especially for long videos.

---

> ### Author Rebuttal · Authors · 2026-03-31
>
> We sincerely thank the reviewer for the constructive comments and for the time and effort devoted to reviewing our manuscript.
>
> > W1: The proposed approach introduce higher latency in time-to-first-frame compared to purely autoregressive generation
>
> Thanks for your insightful question. Your understanding is correct. Compared with a purely AR model with chunk size = 1 frame, our method inevitably incurs a higher time-to-first-frame. If compared with the self-forcing setting with chunk size = 3, this issue can be partially alleviated by introducing a flexible chunking scheduler over both frames and timesteps (a larger chunk size of 4 at the first step, and then switch to a smaller chunk size of 2 for the subsequent steps).
>
> We think our method is not particularly well-suited for interactive scenarios. Instead, it is better suited for the setting of **generating a complete video more efficiently and with higher overall quality**. This is the main reason why we focus on the results in Table 2: compared with the self-forcing and bidirectional model, our method achieves higher FPS while also delivering better video quality.
>
> > W2-1: It is unclear whether the approach works at larger model scales or with different architectures.
>
> Thanks for your great question. We present here the result of applying flex-forcing on SANA-Video:
>
> * Comparison with Self-Forcing on SANA-Video. Our method exceeds the LongSANA trained with self-forcing.
>
> | | Quality|Semantic|Total|
> | - | - | - | - |
> |LongSANA Self-forcing (Official checkpoint) |82.29|77.17|81.27|
> |SANA-Video Flex-Forcing |82.33|79.30|81.72|
>
> Training and evaluation configuration: We evaluate SANA-Video using the official LongSANA-Video-2B-Self-Forcing checkpoints and the official inference configuration. We do not perform the subsequent LongSANA training after self-forcing, so some performance gap is expected compared with models that undergo the full LongSANA training as reported in the paper.
>
> * Flexibility under different chunking strategies:
>
> We further verify that Flex-Forcing preserves its key advantage of flexible chunking at inference time. We evaluate the trained model under multiple chunking strategies. Most of the results surpass the performance of official self-forcing checkpoint of LongSANA (Total score = 81.27)
>
> |Chunk Strategy |Quality|Semantic|Total|
> |-|-|-|-|
> |12-9 | 82.27 | 78.66 | 81.55|
> |13-8 | 82.33 | 78.94 | 81.66|
> |7-7-7 | 82.00 | 78.91 | 81.38|
> |9-6-6 | 82.14 | 79.01 | 81.52|
> |6-5-5-5 | 81.94| 79.11| 81.37|
> |3-3-3-3-3-3-3 | 80.23| 79.58| 80.10|
>
> > W2-2: It would be helpful to include human evaluations
>
> The user study results for both short and long videos are presented in Figure 10 of the manuscript, where we report the win rate of our method against baselines ( self-forcing for 5-second videos and Infinity-ROPE for 30-second videos). The evaluation samples are randomly selected from a pool of approximately 1,000 generated videos.
>
> || 5s video | 30s video |
> |-|-|-|
> |Visual Quality|55.43%|53.93%|
> |Prompt Alignment| 53.26%| 50.56%|
>
> The gain is more pronounced in visual quality than in prompt alignment, which is expected since our method is designed to improve how video content is generated across frames and denoising timesteps, which more directly influences perceptual fidelity and temporal consistency.
>
> > Q1: Whether the proposed hybrid approach provides a clear advantage over a fully bidirectional DM?
>
> Thank you for the great question. Our method provides a better quality-efficiency trade-off than a fully bidirectional diffusion model.
>
> |Method|Diffusion Time (ms)|VAE Time (ms)|FPS|VBench|
> |-|-|-|-|-|
> |Wan with DMD-v, fully bidirectional one |3208|1171|25.25|84.60|
> |Ours - best performance (15-3-3) | 3138 |1168|25.80|85.07|
> |Ours - best speed (7-7-7) |2757|1171|29.37|84.63|
> |Ours - best performance-speed trade-off (9-6-6) |2793|1172|28.99|84.83|
>
> Our method achieves a more favorable balance between generation quality and efficiency. The best-performance setting improves VBench from 84.60 to 85.07, and the best-speed setting further increases throughput to 29.37 FPS while preserving comparable quality. The best quality-efficiency trade-off is achieved by a specific chunk configuration (9-6-6).
>
> > Limitations
>
> Sorry for missing this section. We provide the limitation discussion below and will include it in the revised manuscript.
>
> Our method still has several limitations. Although it relaxes the strict left-to-right training constraint, the training–inference mismatch is not fully resolved and can still accumulate errors in long video generation. In addition, the effectiveness of the method appears to depend heavily on capabilities inherited from pre-training, especially bidirectional encoding priors, which may limit its transferability to models where such priors are weak or absent. More broadly, as with other generative video models, the proposed method may be misused in downstream applications if deployed without appropriate safeguards.

---

> > ### Author Rebuttal · Reviewer_4fb8 · 2026-04-01
> >
> > Thank you for the response and the additional clarifications.
> >
> > The rebuttal helps improve the clarity of the paper and addresses several of my questions. I appreciate the additional experimental evidence and the discussion of limitations.
> >
> > That said, my original concerns regarding the time-to-first frame and the breadth of evaluation are only partially addressed. As a result, my overall assessment remains unchanged.
> >
> > I will keep my original score.

---

### Official Review · Reviewer_VE3s · 2026-03-12

**Soundness:** 2
**Presentation:** 3
**Significance:** 2
**Originality:** 2
**Overall Recommendation:** 4
**Confidence:** 4

**Summary:**

The paper proposes Flex-Forcing, a unified framework for video diffusion models that enables both bidirectional and autoregressive inference through flexible chunking over temporal frames and diffusion timesteps. The method aims to combine the global coherence of bidirectional diffusion models with the efficiency and scalability of autoregressive generation. However, compare with self-forcing style approaches, this work has limited novelty. And theoretical analysis is minimal.

**Compliance With Llm Reviewing Policy:**

Affirmed.

**Final Justification:**

I update the score to "weak accept".

**Key Questions For Authors:**

1. How well does the optimal chunk configuration generalize across: different prompts, different video lengths, and different models?

2. Is the improvement primarily due to temporal chunking, timestep chunking, or their combination? An ablation isolating the contribution of each axis would help clarify the source of the gains.

3. Could the same flexible chunking principle be applied to autoregressive transformer video models or other generative architectures? Clarifying this would help understand whether Flex-Forcing is a diffusion-specific technique or a more general generation paradigm.

**Limitations:**

yes

**Strengths And Weaknesses:**

Strengths:

1. The authors propose an interesting problem in video generation about efficiency vs quality.

2. Simple and clean framework and empirical improvements.

Weaknesses:

1. The authors use brute-force search chunk configurations. It's an ad-hoc strategy and has limited generalization ability.

2. Limited evaluation. The authors do not discuss about different diffusion backbone and  there is no video demo.

3. The authors use bidirectional inside chunk and ar across chunk. It has limited novelty compared with self-forcing / blockwise AR transformers.

---

> ### Author Rebuttal · Authors · 2026-03-31
>
> We sincerely thank the reviewer for the constructive comments and for the time and effort devoted to reviewing our manuscript.
>
> > W1: The brute-force search is an ad-hoc strategy and has limited generalization ability.
>
> We thank the reviewer for the insightful comment. We agree that the current chunk configuration selection is heuristic and relies on brute-force search over a predefined configuration space. This is indeed a limitation of the current paper.
>
> Our primary goal is to study whether flexible chunking is possible and has great benefits, and whether flexible forcing itself can improve video generation by relaxing the strict generation chunking strategy. We view the current brute-force search as a practical instantiation used to expose the potential of the framework, and we show that over 73\% of the configurations would have better performance than the baseline, demonstrating the potential of flexible chunking during inference.
>
> > W2 & Q3: Limited evaluation about different diffusion backbones.  Q1: How well does the optimal chunk configuration generalize?
>
> Thanks for your great suggestion. Here, we show more results if we apply flex-forcing on SANA-Video, which has a different backbone than Wan-2.1. To show the generalizability of the chunk configuration, we directly adopt the best chunk setting on Wan2.1 here.
>
> * Comparison with Self-Forcing on SANA-Video. Our method exceeds the LongSANA trained with self-forcing.
>
> | | Quality|Semantic|Total|
> | - | - | - | - |
> |LongSANA Self-forcing (Official checkpoint) |82.29|77.17|81.27|
> |SANA-Video Flex-Forcing |82.33|79.30|81.72|
>
> Training and evaluation configuration: We evaluate SANA-Video Self-Forcing using the officially released LongSANA-Video-2B-Self-Forcing checkpoints and the official inference configuration. We do not perform the subsequent LongSANA training stage after self-forcing, so some performance gap is expected compared with models that undergo the full LongSANA training pipeline, as reported in the paper.
>
> * Flexibility under different chunking strategies:
>
> We further verify that Flex-Forcing preserves its key advantage of flexible chunking at inference time. We evaluate the trained model under multiple chunking strategies. Most of the results surpass the performance of official self-forcing checkpoint of LongSANA (Total score = 81.27).
>
> |Chunk Strategy |Quality|Semantic|Total|
> |-|-|-|-|
> |12-9 | 82.27 | 78.66 | 81.55|
> |13-8 | 82.33 | 78.94 | 81.66|
> |7-7-7 | 82.00 | 78.91 | 81.38|
> |9-6-6 | 82.14 | 79.01 | 81.52|
> |6-5-5-5 | 81.94| 79.11| 81.37|
> |3-3-3-3-3-3-3 | 80.23| 79.58| 80.10|
>
> > W3: It has limited novelty compared with self-forcing / blockwise AR transformers.
>
> Thanks for your valuable feedback. We agree that our method shares a similar high-level intuition with self-forcing with block-wise autoregressive generation. However, our main contribution is not simply another chunked generation model, but a flexible generation framework in which a single model can support different chunk configurations rather than committing to one fixed chunking schedule.
>
> With this, FlexForcing has the key advantage that
> (1) New editing capability: FlexForcing expands the functional capability of the model by enabling any-order and any-timestep editing, which self-forcing does not naturally support.
> (2) Better performance: a more favorable quality-efficiency trade-off than committing to one fixed schedule.
>
> From this perspective, we view the novelty less as introducing another variant of chunked generation, and more as demonstrating that **flexibility itself can serve as a useful generation paradigm**.
>
> > Q2: Is the improvement primarily due to temporal chunking, timestep chunking, or their combination? An ablation isolating the contribution of each axis would help.
>
> Thank you for this insightful question. The improvement comes from both axes, and they contribute in different ways. We summarize the ablation below:
>
> | Method |  FPS | VBench |
> | - | - | - |
> | Baseline (self-forcing) | 24.9 | 84.03 |
> | + Temporal Chunking | 29.4 | 84.63 |
> | + Timestep Chunking | 26.1 | 85.01 |
>
> We first apply temporal chunking only and keep the same chunk schedule across all denoising timesteps, the FPS improves from 24.9 to 29.4, while the VBench increases from 84.03 to 84.63.
> When we further introduce timestep chunking on top of temporal chunking, the VBench score improves further to 85.0, while the FPS becomes 26.1. This indicates that timestep chunking mainly provides an additional quality gain, although with some reduction in throughput compared with using temporal chunking alone.
>
> > W2: No video demos.
>
> We uploaded some video demos on this anonymous link, showing the comparison between our method and self-forcing. Please check here for reference: https://anonymous.4open.science/r/FlexForcing-FB97/

---

> > ### Author Rebuttal · Reviewer_VE3s · 2026-04-04
> >
> > Thank you for the detailed and helpful rebuttal. The additional experiments and clarifications, especially regarding the comparison with self-forcing and the flexibility of the framework, have improved my understanding of the work. For future versions, I would suggest further clarifying and positioning the differences and connections with self-forcing and blockwise autoregressive approaches, as this would help better highlight the novelty of the method.

---

### Official Review · Reviewer_i77n · 2026-03-12

**Soundness:** 3
**Presentation:** 3
**Significance:** 3
**Originality:** 3
**Overall Recommendation:** 5
**Confidence:** 4

**Summary:**

The authors introduce Flex-forcing, a unified framework designed to operate seamlessly across both bidirectional and autoregressive generation regimes. The core innovation lies in a flexible chunking mechanism applied during both training and inference. By partitioning videos into dynamic chunks whose compositions vary across denoising steps, the model ensures that each noised chunk is conditioned on preceding clean segments. This design enables any-order, any-timestep autoregressive generation, facilitating diverse applications such as hybrid chunking and flexible video editing.

**Compliance With Llm Reviewing Policy:**

Affirmed.

**Final Justification:**

My concerns have been satisfactorily resolved by the rebuttal. Therefore, I raise my score to 5.

**Key Questions For Authors:**

Please refer to the points raised in the Weaknesses section above.

**Limitations:**

No.
The authors should include a brief discussion on the model's limitations, such as computational overhead, and address potential negative societal impacts like the risk of synthetic media misuse.

**Strengths And Weaknesses:**

## Strengths
* **Novel Unified Generative Paradigm**: Flex-forcing successfully bridges the gap between bidirectional and autoregressive generation within a single, cohesive architecture, offering exceptional flexibility across diverse inference strategies.
* **Effective Balancing of Quality and Efficiency**: The framework demonstrates a robust ability to manage the trade-off between visual quality and computational overhead by dynamically adjusting chunk compositions throughout the denoising process.
* **Versatile Editing and Any-order Support**: The model effectively enables hybrid chunking for enhanced generation quality and flexible, arbitrary-order video editing.
* **Solid Empirical Validation of K-projection**: The authors provide convincing evidence for the effectiveness of K-projection, improving the stability and overall quality of the generated video sequences.

## Weaknesses
* **Clarification of Any-Order Conditioning**: To further strengthen the claim of "any-order" generation, it would be highly beneficial to provide a more detailed rationale for how the model supports future-frame conditioning (e.g., Eq. 7) despite being primarily trained on left-to-right flex-chunk modeling. Clarifying this point would prevent any potential concerns regarding overclaiming and solidify the framework's theoretical foundation.
* **Refinement of Notational Clarity**: The manuscript's impact could be improved by using more distinct notations for different chunking strategies—specifically distinguishing between those with consistent vs. varying chunk shapes across denoising steps. Adding such descriptions to figure and table captions would greatly enhance readability and ensure the mechanism is easily reproducible by the community.
* **Strengthening Technical Self-Containment**: To make the paper more accessible to a broader audience, providing a bit more background on the acquisition of $s_{real}$ and $s_{fake}$ in Eq. 4 and the ODE initialization would be very helpful. Including a Preliminary section or an Appendix detailing related paradigms like CausVid or Self-Forcing would ensure the manuscript is fully self-contained and even more impactful for readers unfamiliar with those specific methods.

---

> ### Author Rebuttal · Authors · 2026-03-31
>
> We sincerely thank the reviewer for the constructive comments and for the time and effort devoted to reviewing our manuscript.
>
> > W1:  Clarification of Any-Order Conditioning: To further strengthen the claim of "any-order" generation, it would be highly beneficial to provide a more detailed rationale for how the model supports future-frame conditioning (e.g., Eq. 7) despite being primarily trained on left-to-right flex-chunk modeling.
>
> Thank you for pointing this out. We agree that this point should be clarified more explicitly, and we will revise the manuscript accordingly.
>
> Our training is not strictly left-to-right. In addition to left-to-right modeling, we also include training cases in which the entire video is treated as a single chunk, or a sufficiently large chunk is denoised jointly. Under this setting, the model uses bidirectional attention within the chunk, allowing each frame to attend to both preceding and succeeding frames in the same chunk. Moreover, future frames are not handled by a separate mechanism: the only distinction between past and future frames comes from the relative positional encoding, which provides the temporal-order signal. Because the model is trained with such bidirectional attention, it already learns to interact with tokens at both positive and negative relative positions, including those corresponding to future frames.
>
> > W2:  Refinement of Notational Clarity: The manuscript's impact could be improved by using more distinct notations for different chunking strategies—specifically distinguishing between those with consistent vs. varying chunk shapes across denoising steps.
>
> Thank you for the valuable feedback. We will revise the notation to more clearly distinguish different chunking strategies and ensure consistency with the notation used in the corresponding figures and tables.
>
>
> > W3:  Strengthening Technical Self-Containment: Including a Preliminary section or an Appendix detailing related paradigms like CausVid or Self-Forcing would ensure the manuscript is fully self-contained and even more impactful for readers unfamiliar with those specific methods.
>
> Thank you for the valuable feedback. We will add a short preliminary section for self-forcing in the main text to make the paper more self-contained.
>
> > Limitations
>
> Sorry for missing this section. We provide the limitation discussion below and will include it in the revised manuscript.
>
> Our method still has several limitations. Although it relaxes the strict left-to-right training constraint, the training–inference mismatch is not fully resolved and can still accumulate errors in long video generation. In addition, the effectiveness of the method appears to depend heavily on capabilities inherited from pre-training, especially bidirectional encoding priors, which may limit its transferability to models where such priors are weak or absent. More broadly, as with other generative video models, the proposed method may be misused in downstream applications if deployed without appropriate safeguards.

---

> > ### Author Rebuttal · Reviewer_i77n · 2026-04-02
> >
> > I thank the authors for their thorough rebuttal and for addressing my concerns. Most of the raised issues have been satisfactorily resolved, and I have one remaining question regarding W1.
> >
> > > W1. Clarification on Any-Order Conditioning
> >
> > Based on my understanding, in a chunk-level, the process is defined as left-to-right denoising, where the left chunk should be fully denoised before the right chunk begins denoising. In that sense, it seems to me that Flex-forcing might not support editing a chunk that lies between preceding and subsequent chunks while conditioning on both. Please let me know if I am wrong.
> >
> > I would like to note that even if this functionality is not currently supported, I still consider this to be a solid and valuable contribution, and the question above is raised out of curiosity and for a better understanding of the method's scope. That said, I think it would be beneficial for the authors to clarify the relevant claims in the paper so that readers have a precise understanding of what the method does and does not support.

---

> > > ### Author Response · Authors · 2026-04-02
> > >
> > > Thank you very much for this insightful question. Yes, you are correct that, at the chunk level our current training remains left-to-right denoising, and therefore does not fully cover all any-order generation cases at inference time, which means that there remains a training–inference gap. Empirically, however, we found that the key factor enabling right-to-left denoising is that the model has been exposed during training to future-positioned tokens through RoPE. This appears sufficient to induce a certain degree of right-to-left prediction ability at inference time, even without explicit chunk-level right-to-left training.
> > >
> > > We think that incorporating chunk-level right-to-left generation, or more generally bidirectional chunk ordering, into training could further reduce this gap and potentially improve performance. We sincerely appreciate this valuable suggestion and believe it is a very promising direction for future work.

---

### Decision · Program_Chairs · 2026-04-30

**Decision:**

Accept (spotlight)

**Comment:**

This paper received unanimous positive reviews (2 WA, 2 A) with the reviewers praising the problem and formulation as interesting + effective and the results as solid. The primary weakness of the paper is the limited scale of the experiments, which is unfortunate but an understandable issue. I advocate for acceptance.